# Bayesian multivariate reanalysis of large genetic studies identifies many new associations

**Michael C. Turchin** [1¤], **Matthew Stephens** [1,2]*

**1** Department of Human Genetics, The University of Chicago, Chicago, Illinois, United States of America,
**2** Department of Statistics, The University of Chicago, Chicago, Illinois, United States of America

¤ Current address: Center for Computational Molecular Biology, Department of Ecology and Evolutionary Biology, Brown University, Providence, Rhode Island, United States of America
* mstephens@uchicago.edu

**Data Availability Statement:** All relevant data are within the manuscript and its Supporting Information files.

**Funding:** This work was supported by National Institutes of Health (NIH) Grant R01 HG002585 to

## Abstract

Genome-wide association studies (GWAS) have now been conducted for hundreds of phenotypes of relevance to human health. Many such GWAS involve multiple closely-related phenotypes collected on the same samples. However, the vast majority of these GWAS have been analyzed using simple univariate analyses, which consider one phenotype at a time. This is despite the fact that, at least in simulation experiments, multivariate analyses have been shown to be more powerful at detecting associations. Here, we conduct multivariate association analyses on 13 different publicly-available GWAS datasets that involve multiple closely-related phenotypes. These data include large studies of anthropometric traits (GIANT), plasma lipid traits (GlobalLipids), and red blood cell traits (HaemgenRBC). Our analyses identify many new associations (433 in total across the 13 studies), many of which replicate when follow-up samples are available. Overall, our results demonstrate that multivariate analyses can help make more effective use of data from both existing and future GWAS.

## Author summary

Genome-wide association studies (GWAS) have become a common and powerful tool for identifying significant correlations between markers of genetic variation and physical traits of interest. Often these studies are conducted by comparing genetic variation against single traits one at a time ('univariate'); however, it has previously been shown that it is possible to increase your power to detect significant associations by comparing genetic variation against multiple traits simultaneously ('multivariate'). Despite this apparent increase in power though, researchers still rarely conduct multivariate GWAS, even when studies have multiple traits readily available. Here, we reanalyze 13 previously published GWAS using a multivariate method and find >400 additional associations. Our method makes use of univariate GWAS summary statistics and is available as a software package, thus making it accessible to other researchers interested in conducting the same analyses. We also show, using studies that have multiple releases, that our new associations have high rates of replication. Overall, we argue multivariate approaches in GWAS should no

MS, NIH Grants T32 GM007197, TL1 TR000432, and F31 AI118375 to MCT, and NIH Grant R01 GM118652. The funders had no role in study design, data collection and analysis, decision to publish, or preparation of the manuscript.

**Competing interests:** The authors have declared that no competing interests exist.

longer be overlooked and how, often, there is low-hanging fruit in the form of new associations by running these methods on data already collected.

## Introduction

Genome wide association studies (GWAS) have been widely used to identify genetic factors—particularly single nucleotide polymorphisms (SNPs) and copy number variations (CNVs)—associated with human disease risk and other phenotypes of interest [1, 2]. Indeed, at time of writing over 24,000 such associations have been identified as 'genome-wide significant' [3].

The vast majority of these many genetic association analyses consider only one phenotype at a time ("univariate association analysis"). This is despite the fact that measurements on multiple phenotypes are often available, and joint association analysis of multiple phenotypes ("multivariate association analysis") can substantially increase power [4–8]. There are likely multiple reasons for the preponderance of univariate analyses. One possible reason is that initial association analyses are usually performed under tight time constraints, and at a time when many other analysis issues (e.g. quality control, population stratification) are competing for attention. In these conditions it makes sense to focus on the simplest possible approach that will quickly yield new associations, without overly worrying about loss of efficiency. In addition analysts may be legitimately concerned that deviation from the most widely adopted analysis pipeline may invite unwanted additional reviewer attention.

Nonetheless, we believe that multivariate association analysis has an important role to play in making the most of costly and time-consuming GWAS studies. One way forward is to conduct multivariate analyses of previously-published GWAS, checking for additional associations that may have been missed by the initial univariate association analyses. This is greatly facilitated by the fact that many GWAS now make summary data from single-SNP tests freely available [9–13], and that simple multivariate analysis can be conducted using such summary data [14–16].

Here we demonstrate the potential benefits of reanalyzing published GWAS using multivariate methods. Specifically we apply multivariate methods from [14] to reanalyze 13 different GWAS whose initial publications reported only univariate results. In most cases our multivariate analyses find many new associations. For example, in GIANT 2014/5 we find over 150 new associations. In studies with multiple data releases, we find that new multivariate associations found in initial releases typically replicate in subsequent releases, supporting that many of the new associations are likely real. We also demonstrate that the multivariate approach is not equivalent to simply relaxing the univariate GWAS significance threshold. Finally, we point out some limitations of the specific framework we used here, and suggest some alternative strategies that may help address those limitations in future multivariate GWAS analyses.

## Results

### Multivariate association analyses

To facilitate multivariate association analyses using the methods from [14], we implemented them in an R package `bmass` (Bayesian multivariate analysis of summary statistics). The software requires as input univariate GWAS summary statistics, for the same set of SNPs, on $d$ related phenotypes. (The derivations in [14] are for quantitative phenotypes, but the methods can also be applied to summary data from binary phenotypes, which can be interpreted as making a normal approximate to the likelihood for the effect sizes as in [17].) Then, for each

SNP, it attempts to categorize each phenotype as belonging to one of three categories: **U**nassociated, **D**irectly Associated, or **I**ndirectly Associated with the SNP. The difference between **D** and **I** is that an indirect association disappears after controlling for associations with other phenotypes (see Methods and S1 Fig).

For $d$ phenotypes, there are $3^d$ possible assignments of phenotypes to these 3 categories, and each assignment corresponds to a different "model" $\gamma$. For example, one model corresponds to the "null" that all phenotypes are **U**nassociated; another model corresponds to the model that all phenotypes are **D**irectly associated; another model corresponds to just the first phenotype being **D**irectly associated, etc. The goal of the association analysis is to determine which of these models is consistent with the data and, in particular, to assess overall evidence against the null model.

The support in the data for model $\gamma$, relative to the null model, is summarized by a Bayes Factor ($\text{BF}_\gamma$). Large values of $\text{BF}_\gamma$ indicate strong evidence for model $\gamma$ compared against the null. One advantage of Bayes Factors over $p$-values is that the Bayes Factors from different models can be easily compared and combined. For example, the overall evidence against the null is given by the (weighted) average of these BFs:

$$\text{BF}_{\text{av}} := \sum_{\gamma} w_\gamma \text{BF}_\gamma \tag{1}$$

where the weights $w_\gamma$ are chosen to reflect the relative plausibility of each model $\gamma$. In `bmass` we implemented the Empirical Bayes approach from [14] that learns appropriate weights from the data (see Methods).

## Comparisons with published univariate analyses

To provide a benchmark against which to compare our multivariate analysis results, we compiled a list of "previous univariate associations": SNPs that were both reported as significant in the original publication and exceeded the original publication's definition for genome-wide significance in at least one phenotype in the publicly-available (univariate) summary data analyzed here. This does not include all SNPs reported in every original publication because in some studies SNPs became genome-wide significant only after adding additional samples not included in the publicly available summary data.

We used these previous univariate associations to determine a significance threshold for our multivariate associations. Specifically, we declared a multivariate association as significant if its $\text{BF}_{\text{av}}$ exceeds that of any previous univariate association's $\text{BF}_{\text{av}}$ in the same study [14]. The rationale is that the evidence for these multivariate associations exceeds the evidence for previously-reported genome-wide significant associations, which are generally regarded as likely to be (mostly) real associations.

Finally, we defined a list of "new multivariate associations", which are SNPs that are significant in our multivariate analysis but are not a "previous univariate association". To avoid double-counting of signals due to linkage disequilibrium (LD), we pruned the list of new multivariate associations so that they are all at least 0.5Mb apart. For additional details, see Methods.

## Many new loci identified in reanalyzing 13 publicly available GWAS studies

We applied `bmass` to 13 publicly available GWAS studies, representing 10 different collections of phenotypes (Table 1). Phenotypic collections include blood lipid traits (GlobalLipids: [9, 18]), body morphological traits (GIANT: [10–12, 19–21]), red blood cell traits

**Table 1. Dataset summary.**

| Dataset | Release | N | Phenotypes |
|---|---|---|---|
| GlobalLipids | 2010 | 95454 | LDL, HDL, TC, TG[a] |
| | 2013 | 188577 | LDL, HDL, TC, TG |
| GIANT | 2010 | 77167 | Height, BMI, WHRadjBMI[b] |
| | 2014/5 | 224459 | Height, BMI, WHRadjBMI |
| HaemgenRBC | 2012 | 135367 | RBC, PCV, MCV, MCH, MCHC, Hb[c] |
| | 2016 | 173480 | RBC, PCV, MCV, MCH, MCHC, Hb |
| ICBP | 2011 | 69395 | SBP, DBP, PP, MAP[d] |
| MAGIC | 2010 | 46186 | FstIns, FstGlu, HOMA_B, HOMA_IR[e] |
| GEFOS | 2015 | 32965 | FA, FN, LS[f] |
| GIS | 2014 | 48972 | Iron, Sat, TrnsFrn, Log10Frtn[g] |
| SSGAC | 2016 | 343072 | NEB_Pooled, AFB_Pooled[h] |
| CKDGen | 2010/1 | 67093 | Crea, Cys, CKD, UACR, MA[i] |
| ENIGMA2 | 2015 | 30717 | ICV, Accumbens, Amygdala, Caudate, Hippocampus, Pallidum, Putamen, Thalamus[j] |

N is the maximum number of samples contributing to each study.

[a]—Low-Density Lipoproteins (LDL), High-Density Lipoproteins (HDL), Total Cholesterol (TC), Total Triglycerides (TG)

[b]—Body Mass Index (BMI), Waist-Hip Ratio adjusted for BMI (WHRadjBMI)

[c]—Red Blood Cell Count (RBC), Packed Cell Volume (PCV), Mean Cell Volume (MCV), Mean Cell Haemoglobin (MCH), Mean Cell Haemoglobin Concentration (MCHC), Haemoglobin (Hb)

[d]—Systolic Blood Pressure (SBP), Diastolic Blood Pressure (DBP), Pulse Pressure (PP), Mean Arterial Pressure (MAP)

[e]—Fasting Insulin (FstIns), Fasting Glucose (FstGlu), Homeostatic Model Assessment of Beta Cell Function (HOMA_B), Homeostatic Model Assessment of Insulin Resistance Function (HOMA_IR)

[f]—Forearm Bone Mineral Density (FA), Femoral Neck Bone Mineral Density (FN), Lumbar Spine Bone Mineral Density (LS)

[g]—Serum Iron (Iron), Serum Transferrin Saturation (Sat), Serum Transferrin (TrnsFrn), Log-Transformed Ferritin (Log10Frtn)

[h]—Number of Children Ever Born, Male & Female (NEB_Pooled), Age at First Birth, Male & Female (AFB_Pooled)

[i]—Serum Creatine (Crea), Serum Cystatin (Cys), Chronic Kidney Disease (CKD), Urinary Albumin-to-Creatine Ratio (UACR), Microalbuminuria (MA)

[j]—Intracranial Volume (ICV), specified subcortical brain structures refer to MRI-derived volume measurements for each one

(HaemgenRBC: [13, 22]), blood pressure traits [23, 24], bone density traits [25], and kidney function traits [26, 27]. For three of these phenotypic collections (GlobalLipids, GIANT, and HaemgenRBC), two different releases were available from the source consortiums. We conducted basic QC as described in Methods.

Our multivariate analyses identify, in total, hundreds of new associations. The numbers of previous univariate associations and new multivariate associations are summarized in Fig 1 (see also Table 2). For example, we identify 162 new multivariate associations in GIANT2014/5, 65 in GlobalLipids2013, and 60 in HaemgenRBC2016. These represent power increases from 10% to 45% compared with previous univariate analyses.

## Replication of multivariate associations across releases

To demonstrate that many of these new multivariate associations are likely to be real we take advantage of three datasets that each have two releases separated by several years (GlobalLipids, GIANT, and HaemgenRBC). In each case we performed multivariate association analysis of the earlier release and checked how the new multivariate associations fared in univariate analyses of the later release (Fig 2). Since later releases include the samples from earlier releases, to assess "replication" we focus on whether the association in the new release is more significant than the original release—that is, whether the signal in the new (non-overlapping)

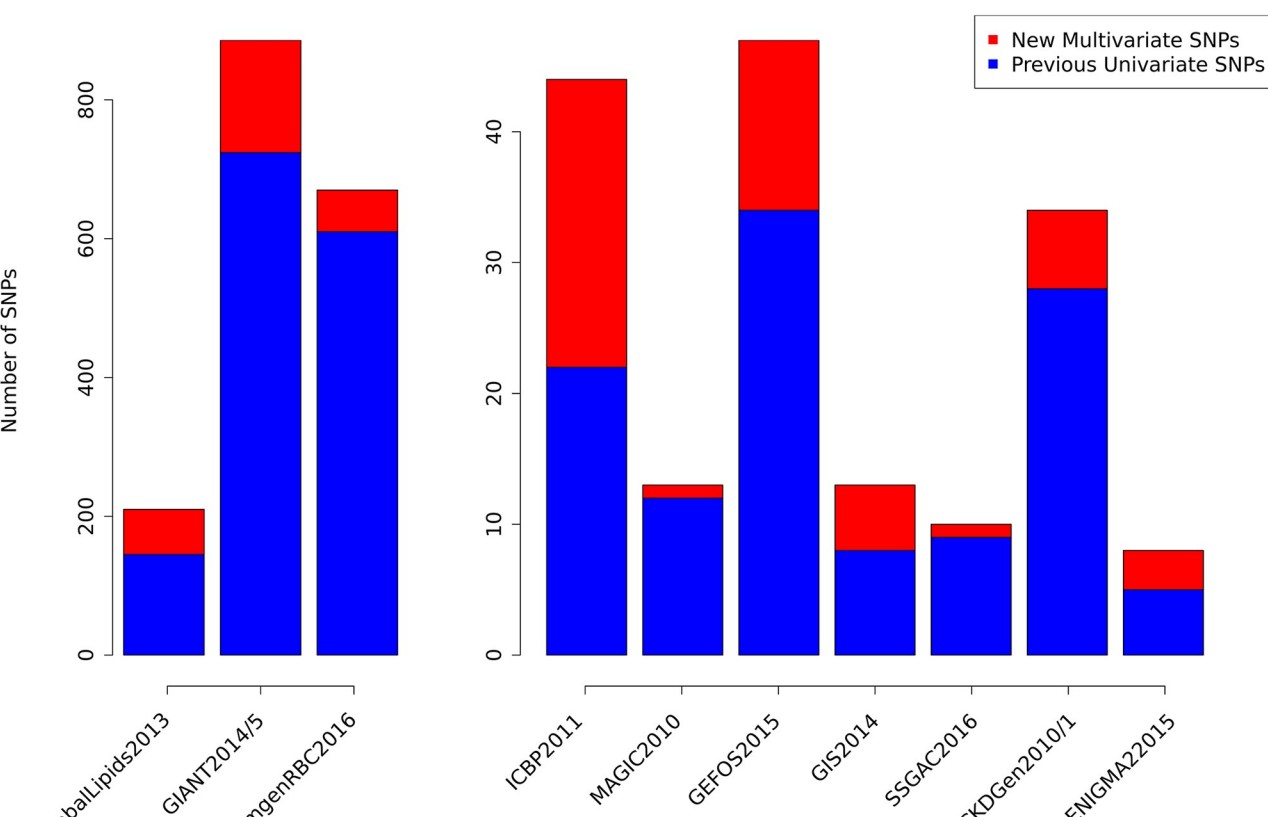

**Fig 1. Number of independent significant SNPs, by study.** The barplot shows the number of independent SNPs that were significant in previous univariate analyses (blue) and the number of additional significant associations in our new multivariate analyses (red). For univariate analysis, significance levels were set by the original study. For multivariate analyses, we declared a SNP to be significant if its weighted average Bayes Factor ($BF_{av}$) exceeded that of the smallest $BF_{av}$ among the previous univariate significant SNPs. We considered SNPs more than.5Mb apart to be independent. See Table 1 and Methods for phenotype details, Methods for further analysis details, and S2–S4 Tables for lists of significant SNPs from each dataset.

samples provides additional evidence *over and above* the original signal. By this measure the results show high replication rates for the new multivariate associations: in total, 84 of 94 new associations have smaller minimum univariate *p*-values in the later release (at exactly the same SNP), and indeed the majority of these reach univariate GWAS significance in the later release.

## Multivariate analysis is different from multiple univariate analyses

Because multivariate analysis takes account of *joint* patterns across phenotypes, its ranking of significance of SNPs can change compared with that from the univariate *p*-values alone. That is, multivariate analysis is not simply equivalent to multiple univariate analyses. To illustrate this we examined, in three well-powered studies, the associations that would be declared significant if the univariate significance threshold were relaxed, and assessed which of them would also be significant in our multivariate analysis (i.e. we assess whether, if we go deeper into the univariate results, we find the same SNPs as appear in our multivariate results). The results are shown in Fig 3. Although there is, understandably, substantial overlap between the

**Table 2. Summary of new multivariate associations identified.**

| | | —SNP Associations— | | | |
|---|---|---|---|---|---|
| Dataset | Release | Previous Univariate | New Multivariate | BF$_{av}$ Thresh | Overlap With Next Release |
| GlobalLipids | 2010 | 102 | 19 | 4.35 | 13/19 |
| | 2013 | 145 | 65 | 4.29 | - |
| GIANT | 2010 | 144 | 60 | 4.11 | 49/60 |
| | 2014/5 | 724 | 162 | 4.49 | - |
| HaemgenRBC | 2012 | 63 | 16 | 5.21 | 9/16 |
| | 2016 | 610 | 60 | 4.68 | - |
| ICBP | 2011 | 22 | 22 | 5.24 | - |
| MAGIC | 2010 | 12 | 1 | 6.90 | - |
| GEFOS | 2015 | 34 | 13 | 5.06 | - |
| GIS | 2014 | 8 | 5 | 7.04 | - |
| SSGAC | 2016 | 9 | 1 | 5.43 | - |
| CKDGen | 2010/1 | 28 | 6 | 4.10 | - |
| ENIGMA2 | 2015 | 5 | 3 | 7.48 | - |

Previous Univariate: the number of previous genome-wide significant univariate associations based on the publicly available summary data. New Multivariate: the number of new genome-wide significant multivariate associations. BF$_{av}$ Thresh: the Bayes Factor threshold used in declaring new multivariate associations to be significant. Overlap With Next Release: for GlobalLipids2010, GIANT2010, and HaemgenRBC2012, the last column shows the number of new multivariate associations that overlap with the univariate GWAS associations in the next release from the same consortium; overlap is defined as being within 50kb of the univariate GWAS variant.

significant SNPs, any non-trivial relaxation of the univariate threshold includes many SNPs that are not multivariate significant in our analysis; for example, at a univariate threshold of $5 \times 10^{-7}$ only 66% of the univariate significant SNPs are also multivariate significant across these three studies. This demonstrates that, indeed, our multivariate approach reorders significance of SNPs compared with multiple univariate analyses.

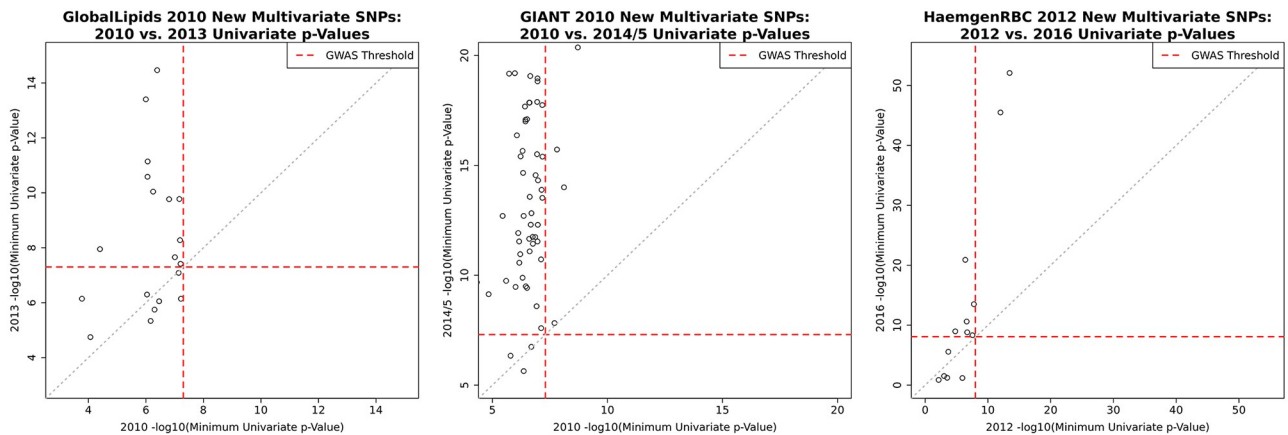

**Fig 2. Replication of new multivariate associations.** The figure shows results based on earlier and later releases from studies with multiple releases (GlobalLipids, GIANT, and HaemgenRBC). Each point represents a new multivariate association identified in our multivariate analysis of the earlier release. The x- and y-axes show the minimum (across phenotypes) of the -log$_{10}$ univariate p-values from the earlier release (x-axis) vs. the later release (y-axis). Dashed red lines represent the univariate significance GWAS thresholds used for each study's releases. Across all three studies, 84 out of 94 new multivariate associations from the earlier releases have smaller minimum univariate p-values in the later release, and 68 out of 84 new multivariate associations that did not reach GWAS significance in the earlier release do so in the later release (see S5 Table for a per-dataset breakdown).

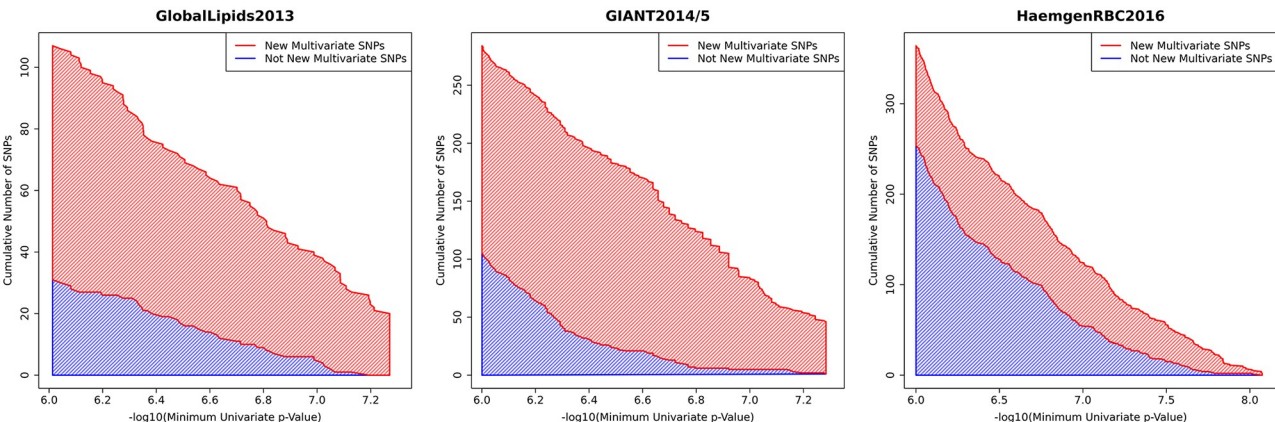

**Fig 3. Comparison of new multivariate hits vs. relaxing univariate *p*-value threshold.** For each data set the graph shows how many associations become significant as the univariate *p*-value threshold is relaxed (moving from right to left on the *x*-axis), and how many of these are declared as new multivariate hits in our analysis. In both cases results are pruned to avoid counting associations of SNPs in strong LD; see Methods for details. The appearance of appreciable blue areas indicates that the multivariate analysis is reordering the significance of SNPs compared with performing multiple univariate analyses.

## Reanalysis also identifies new univariate associations

During our multivariate reanalyses we noticed many SNPs that appeared to be genome-wide univariate significant but were—somewhat mysteriously—not reported as such by the original studies (i.e. SNPs whose univariate *p*-values crossed the significance threshold, as defined by the given study, in at least one trait). S1 Table reports 79 such associations.

There may be many reasons why such variants went unreported, but one reason may be physical proximity to a variant with a stronger signal. Indeed, more than half of the variants described above are within 1Mb of a previously-reported univariate GWAS association. Refraining from reporting multiple near-by associations seems a reasonable—if conservative—strategy to avoid reporting redundant associations due to LD. Further, even when redundant associations due to LD can be ruled out (e.g. by directly examining LD rather than by simply using physical distance), it might be assumed that multiple nearby associated variants may all act through the same biological mechanism and therefore provide redundant biological insights. However, we found that multi-phenotype patterns of association can differ between nearby SNPs, suggesting that they act through different mechanisms.

To highlight just one example, consider rs7515577—which is an original univariate association in GlobalLipids2010—and rs12038699—which is a new multivariate association in GlobalLipids2013. We note that rs12038699 actually reached univariate genome-wide significance in the GlobalLipids2013 dataset, but was not reported (S6 Table). This is possibly because the latter SNP is relatively close, in genomic terms, to the former SNP (549kb). However, these SNPs are not in strong LD ($r^2 = .08$), and so these associations almost certainly represent non-redundant associations. This is further supported by the effect sizes in each phenotype, which clearly reveal very different multivariate patterns of effect sizes among phenotypes (S2 Fig & S6 Table). Indeed the very different multivariate patterns of effect size suggest that not only are these associations non-redundant but likely involve different biological mechanisms as well.

These results suggest that, moving forward, it may pay to be more careful in designing filters designed to avoid reporting redundant associations, and that multi-phenotype analyses may have a helpful role to play here.

## Limitations

One goal of the multivariate approach introduced in [14] was to increase interpretability of multivariate analyses; in particular, the goal was to not only *test* for associations but also to help *explain* the associations by partitioning the phenotypes into "Unassociated", "Directly Associated", and "Indirectly Associated" categories. In principle one might hope to use these classifications to gain insights into the relationships among phenotypes and also perhaps to identify different "types" of multivariate association—effectively clustering associations into different groups. However, in practice we find that these discrete classifications are often not as helpful as one might hope. One reason is the difficulty of reliably distinguishing between direct and indirect effects [14]. Another reason is widespread associations with multiple phenotypes. Indeed, we find that, consistently across data sets, the most common multivariate models involve associations—either direct or indirect—with many phenotypes (S7 Table) and many SNPs are classified as being associated with many phenotypes (Fig 4A). Further, SNPs are very rarely confidently classified as "Unassociated" with any phenotype (Fig 4B). This last observation can be explained by the fact that it is essentially impossible to distinguish

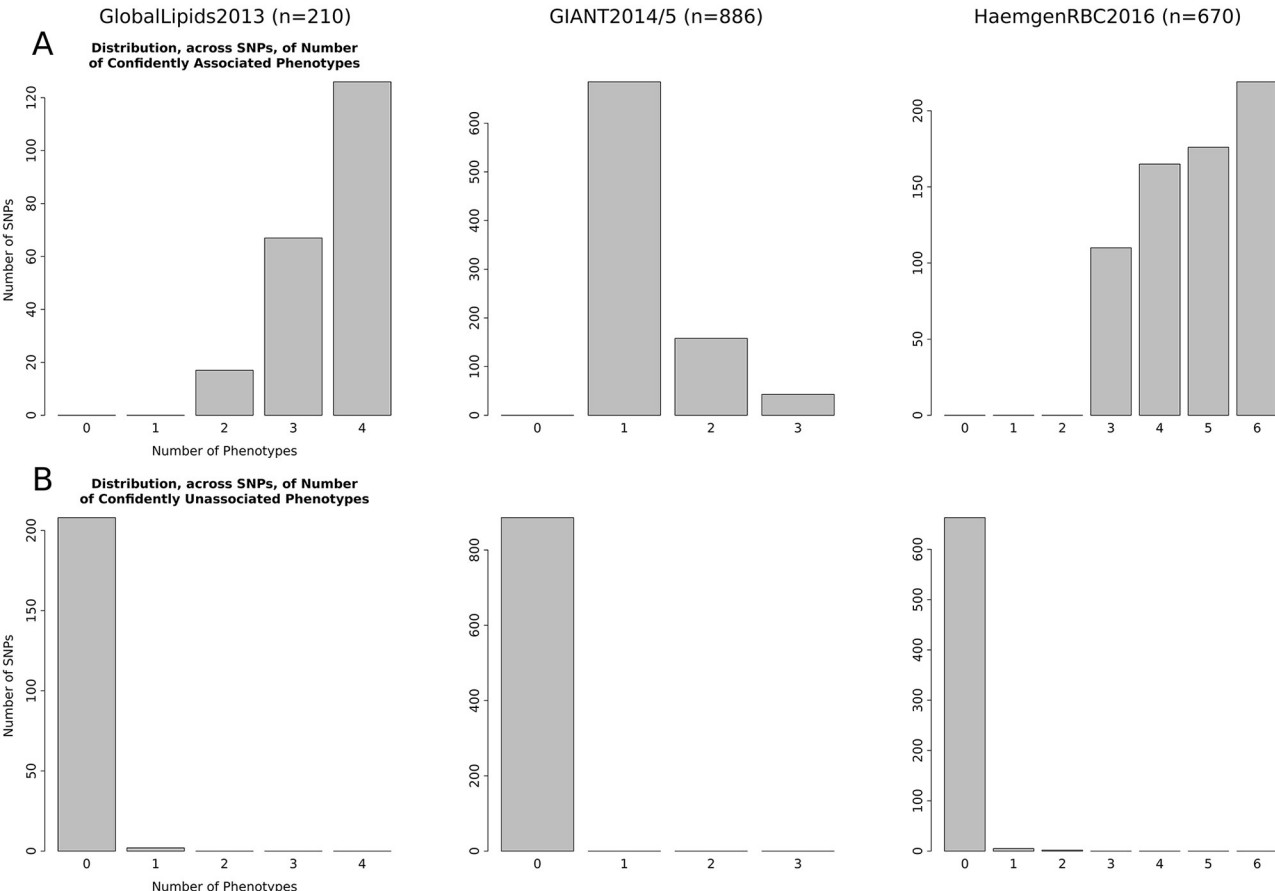

**Fig 4. Distribution, across significant SNPs, of number of phenotypes that are confidently associated (A) or confidently unassociated (B).** Results are shown for three well-powered datasets: GlobalLipids2013, GIANT2014/5, and HaemgenRBC2016. Here "confident" means with probability >0.95, so a SNP is considered "confidently associated" with a phenotype if the sum of its probabilities in the "Directly Associated" and "Indirectly Associated" categories exceeds 0.95 (A), and is considered confidently unassociated with the phenotype if this probability is less than 0.05 (B). The set of significant SNPs includes both previous univariate associations and new multivariate associations.

'unassociated' from 'weakly associated'. Nonetheless when all SNPs show similar classifications it is difficult to get insights into different patterns of multivariate association.

Moving forward, rather than relying on the discrete classifications of "Unassociated", "Directly Associated", and "Indirectly Associated" to identify different patterns of multivariate association, we believe it will be more fruitful to use multivariate methods that take a more quantitative approach, such as identifying different patterns of *effect size* (including direction of effect) among phenotypes [28]. Focusing on effect sizes has the potential to be much more informative than discrete classification, which can hide effect size differences. For example, when multiple SNPs are classified as associated with all phenotypes, they can still show very different patterns of estimated effect sizes/direction (see S3 Fig).

Another limitation of our multivariate methods is that they can lead to (what appear to be) false positive associations when applied to test SNPs with very low minor allele frequencies. Specifically we saw examples where very low-frequency SNPs (e.g. MAF < .001) showed strong signals of multivariate association despite showing very little signal in any univariate test. Although such results are not impossible, we believe that most of these cases were likely false positives, and we applied a MAF cut-off (of 0.01 or 0.005) to avoid these issues. Consequently we recommend caution in interpreting results of multivariate analyses at very low-frequency SNPs, and more generally we recommend that multivariate results be compared against univariate results to check they make sense—highly significant multivariate associations that do not also show at least a moderate level of univariate association should be treated with caution.

## Discussion

We reanalyzed 13 publicly available GWAS datasets using a Bayesian multivariate approach and identified many new genetic associations. Turning genetic associations into biological discoveries remains, of course, a challenging problem. Nonetheless, our results suggest that the increased power of multivariate association analysis that has been reported in many simulation studies [8, 14, 29] also translates to discovery of many new associations in practice.

Our results exploit the public availability of summary data from several large GWAS. Despite progress toward easier availability of individual-level data for large studies [30], in many cases summary data remain much easier to obtain and work with; there are big practical advantages as well to modular pipelines that first compute summary data and then use these as inputs to subsequent (more sophisticated) analyses. For example, the multivariate analyses we present here are simplified by assuming that the summary data were computed while adequately adjusting for population stratification and other relevant covariates (indeed, our current `bmass` software implementation does not allow adjustment for covariates, and so any adjustment must be done in the univariate analyses). And our results illustrate the potential for reanalysis of summary data to yield novel inferences. In this regard we also emphasize the importance of consortia releasing carefully-chosen summaries. For example, $Z$-scores are much more helpful than $p$-values because they preserve information on the direction of the effect. Even better would be both the effect size and standard error that created the $Z$-score. More generally, it is always helpful to include additional key meta-data (e.g. the reference allele, or effect allele, the minor allele frequency, and sample size).

The specific multivariate methods used here were derived under the assumption that the summary data from each phenotype has been obtained from the same sampled individuals (which is true, at least approximately, for studies analyzed here). However, multivariate analysis of summary data is also possible even when data were obtained from different samples for each phenotype. The main difference between these settings is that, for data from overlapping

samples, the "noise" is correlated as well as the signal: i.e. the summary data are correlated under the null due to sample overlap, and correlated under the alternative due to both sample overlap and any shared genetic effects. In contrast, for data from non-overlapping samples the noise is uncorrelated (whereas the signal may remain correlated if genetic factors are shared). Our methods use data at (empirically) null SNPs to estimate the noise correlation, and so their overall assessment of associations should be relatively robust to whether samples for different phenotypes overlap (however, our definitions of **D** (direct) vs **I** (indirect) associations requires the same samples to be measured across phenotypes.)

Moving forward, we expect multivariate association analyses to play an increasingly important role in detecting and understanding genetic associations and relationships among phenotypes. Large studies are now collecting, and making available, rich human genetic and phenotypic information on many complex phenotypes, most notably the UKBioBank [30]. In addition, there are increasingly large studies linking genetic variation and molecular phenotypes such as gene expression (e.g. the GTEx project [31]), as well as epigenetic modifications and transcript degradation [32–35]. Analysis of multiple molecular traits can help yield insights into causal connections among traits [36], and joint analysis of molecular traits with complex phenotypes may also shed light on functional mechanisms (as in "co-localization" analyses [16, 37–39]). Even simply moving from single phenotype to pairwise analysis can shed considerable light on sharing of genetic effects and possible causal connections [15, 40].

These increasingly-complex new data also bring new analytic and computational challenges. Here we have restricted our analyses to relatively small sets of closely-related traits, and indeed the specific multivariate framework we used here—which performs an exhaustive search over all possible multivariate models—is fully tractable for only moderate numbers of traits (up to about 10). Scaling methods up to dealing with larger number of traits may well be helpful for some settings, and recent multivariate analysis methods can deal with dozens of outcomes [28, 41]. In addition, developing multivariate methods to perform *fine-mapping* of genetic associations simultaneously across multiple phenotypes [42] seems an important and challenging area for future work.

## URLs

`bmass` R package: https://github.com/mturchin20/bmass.

## Methods

### GWAS datasets

Below are specific details regarding retrieval and data-processing for each dataset analyzed. Where applicable, these details include the sample size (*N*), minor allele frequency (MAF), and *p*-value thresholds that were applied (based on the thresholds used in the original publications). For each dataset variants were dropped if they satisfied at least one of the following criteria: did not contain information for every phenotype; had missing MAF; were fixed (MAF of 0); had effect size exactly 0 (i.e. direction of effect would be indeterminable); or did not contain the same reference and alternative alleles across each phenotype. For a handful of studies, external databases were used to retrieve chromosome, basepair information, and MAF based on rsID#; in these studies SNPs for which this information could not be retrieved were also dropped.

**GlobalLipids2010** [18]: Original merged, processed, and GWAS-hit annotated summary data from [14] for HDL, LDL, TG, and TC was downloaded from https://github.com/stephens999/multivariate (*dtlesssignif.annot.txt* and *RSS0.txt*).

**GlobalLipids2013** [9]: Summary data for HDL, LDL, TG, and TC was downloaded from http://csg.sph.umich.edu/abecasis/public/lipids2013/. We used a minimum $N$ threshold of 50,000, a MAF threshold of 1%, and a univariate significant GWAS $p$-value threshold of $5 \times 10^{-8}$. All variants were oriented to the HDL minor allele. The final merged and QC'd datafile contained 2,004,701 SNPs. rsID#'s of published GWAS SNPs were retrieved for all four phenotypes from https://www.nature.com/ng/journal/v45/n11/full/ng.2797.html via Supplementary Tables 2 and 3.

**GIANT2010** [19–21]: Summary data for Height, BMI, and WHRadjBMI were downloaded from https://www.broadinstitute.org/collaboration/giant/index.php/GIANT_consortium_data_files. We used a minimum $N$ threshold of 50,000, a MAF threshold of 1%, and a univariate significant GWAS $p$-value threshold of $5 \times 10^{-8}$. Chromosome and basepair position per variant were retrieved from dbSNP130 [43]. All variants were oriented to the Height minor allele. The final merged and QC'ed datafile contained 2,363,881 SNPs. rsID#'s of published GWAS SNPs were retrieved for Height from https://www.nature.com/nature/journal/v467/n7317/full/nature09410.html via Supplementary Table 1, for BMI from https://www.nature.com/ng/journal/v42/n11/full/ng.686.html via Table 1, and for WHRadjBMI from https://www.nature.com/ng/journal/v42/n11/full/ng.685.html via Table 1.

**GIANT2014/5** [10–12]: Summary data for Height, BMI, and WHRadjBMI were downloaded from https://www.broadinstitute.org/collaboration/giant/index.php/GIANT_consortium_data_files. We used a minimum $N$ threshold of 50,000, a MAF threshold of 1%, and a univariate significant GWAS $p$-value threshold of $5 \times 10^{-8}$. Chromosome and basepair position per variant were retrieved from dbSNP130 [43]. All variants were oriented to the Height minor allele. The final merged and QC'ed datafile contained 2,340,715 SNPs. rsID#'s of published GWAS SNPs were retrieved for Height from https://www.nature.com/ng/journal/v46/n11/full/ng.3097.html via Supplementary Table 1, for BMI from https://www.nature.com/nature/journal/v518/n7538/full/nature14177.html via Supplementary Tables 1 and 2, and for WHRadjBMI from https://www.nature.com/nature/journal/v518/n7538/full/nature14132.html via Supplementary Table 4.

**HaemgenRBC2012** [22]: Summary data for RBC, PCV, MCV, MCH, MCHC, and Hb were downloaded from the European Genome-Phenome Archive via accession number EGAS00000000132 (https://www.ebi.ac.uk/ega/studies/EGAS00000000132). We used a minimum $N$ threshold of 10,000, a MAF threshold of 1%, and a univariate significant GWAS $p$-value threshold of $1 \times 10^{-8}$. Chromosome, basepair position, and MAF per variant were retrieved from HapMap release 22 [44]. All variants were oriented to the RBC minor allele. The final merged and QC'ed datafile contained 2,327,567 SNPs. rsID#'s of published GWAS SNPs were retrieved for all six phenotypes from https://www.nature.com/nature/journal/v492/n7429/full/nature11677.html via Table 1.

**HaemgenRBC2016** [13]: Summary data for RBC, PCV, MCV, MCH, MCHC, and Hb were shared via personal communication with the authors. We used a MAF threshold of 1% and a univariate significant GWAS $p$-value threshold of $8.319 \times 10^{-9}$. Since sample size was not provided per variant, the following overall study sample sizes were used as proxies per phenotype: 172,952 for RBC, 172,433 for PCV, 173,039 for MCV, 172,332 for MCH, for 172,925 MCHC, and 172,851 for Hb. All variants were oriented to the RBC minor allele. Only SNPs were analyzed. The final merged and QC'ed datafile contained 8,649,095 SNPs. We then used these summary data to create a list of (non-redundant) "Previous univariate associations". This was done separately for each phenotype by collecting all SNPs that exceeded the univariate significant GWAS $p$-value threshold and greedily pruning the SNPs: i.e. we went down the list, removing SNPs that were less significant than another SNP within 500kb. The pruned lists of previous univariate associations for each phenotype were then combined to produce the final

SNP list of "published GWAS results". Published CNVs that tagged regions that were not identified by this 'final published SNP list' were also included to avoid erroneously claiming downstream a region as a 'new unpublished result'; these CNVs however were only used to mask additional loci as being 'nearby a published univariate GWAS result' and for nothing else in the bmass analysis pipeline.

**ICBP2011** [23, 24]: Summary data for SBP, DBP, PP, and MAP were downloaded from dbGaP via accession number phs000585.v1.p1 (https://www.ncbi.nlm.nih.gov/projects/gap/cgi-bin/study.cgi?study_id=phs000585.v1.p1). We used a minimum $N$ threshold of 10,000, a MAF threshold of 1%, and a univariate significant GWAS $p$-value threshold of $5 \times 10^{-8}$. Chromosome and basepair position per variant were retrieved from HapMap release 21 [44]. All variants were oriented to the SBP minor allele. The final merged and QC'ed datafile contained 2,387,851 SNPs. rsID#'s of published GWAS SNPs were retrieved for SBP and DBP from https://www.nature.com/nature/journal/v478/n7367/full/nature10405.html via Supplementary Table 5, and for PP and MAP from https://www.nature.com/ng/journal/v43/n10/full/ng.922.html via Table 1 and Supplementary Table 2F. Additionally, we gratefully acknowledge the International Consortium for Blood Pressure Genome-Wide Association Studies (Nature. 2011 Sep 11;478(7367):103-9, Nat Genet. 2011 Sep 11;43(10):1005-11) for generating and sharing these data.

**MAGIC2010** [45]: Summary data for FstIns, FstGlu, HOMA_B, and HOMA_IR were downloaded from https://www.magicinvestigators.org/downloads/. We used a MAF threshold of 1% and a univariate significant GWAS $p$-value threshold of $5 \times 10^{-8}$. Since sample size was not provided per variant, the overall study sample size of 46,186 was used as a proxy. Chromosome and basepair position per variant were retrieved from HapMap release 22 [44]. All variants were oriented to the FstIns minor allele. The final merged and QC'ed datafile contained 2,333,328 SNPs. rsID#'s of published GWAS SNPs were retrieved for all four phenotypes from https://www.nature.com/ng/journal/v42/n2/full/ng.520.html via Table 1.

**GEFOS2015** [25]: Summary data for FA, FN, and LS were downloaded from http://www.gefos.org/?q=content/data-release-2015. We used a MAF threshold of.5% and a univariate significant GWAS $p$-value threshold of $1.2 \times 10^{-8}$. Since sample size was not provided per variant, the overall study sample size of 32,965 was used as a proxy. All variants were oriented to the FA minor allele. The final merged and QC'ed datafile contained 8,938,035 SNPs. rsID#'s of published GWAS SNPs were retrieved for all four phenotypes from https://www.nature.com/nature/journal/v526/n7571/full/nature14878.html via Supplementary Table 13.

**GIS2014** [46]: Summary data for Iron, Sat, TrnsFrn, and Log10Frtn were shared via personal communication with the authors. We used a MAF threshold of 1% and a univariate significant GWAS $p$-value threshold of $5 \times 10^{-8}$. Since sample size was not provided per variant, the overall study sample size of 48,972 was used as a proxy. All variants were oriented to the Iron minor allele. The final merged and QC'ed datafile contained 1,985,313 SNPs. rsID#'s of published GWAS SNPs were retrieved for all four phenotypes from https://www.nature.com/articles/ncomms5926/ via Table 1.

**SSGAC2016** [47]: Summary data for NEB_Pooled and AFB_Pooled were downloaded from https://www.thessgac.org/data. We used a MAF threshold of 1% and a univariate significant GWAS $p$-value threshold of $5 \times 10^{-8}$. Since sample size was not provided per variant, the following overall study sample sizes were used as proxies per phenotype: 251,151 for NEB_Pooled and 343,072 for AFB_Pooled. All variants were oriented to the NEB_Pooled minor allele. The final merged and QC'ed datafile contained 2,395,561 SNPs. rsID#'s of published GWAS SNPs were retrieved for all four phenotypes from https://www.nature.com/ng/journal/v48/n12/full/ng.3698.html via Table 1.

**CKDGen2010/1** [26, 27]: Summary data for Crea, Cys, CKD, UACR, and MA were downloaded from https://www.nhlbi.nih.gov/research/intramural/researchers/pi/fox-caroline/datasets. We used a MAF threshold of 1% and a univariate significant GWAS $p$-value threshold of $5 \times 10^{-8}$. Since sample size was not provided per variant, the following overall study sample sizes were used as proxies per phenotype: 67,093 for Crea, 20,957 for Cys, 62,237 for CKD, 31,580 for UACR, and 30,482 for MA. All variants were oriented to the Crea minor allele. The final merged and QC'ed datafile contained 2,333,498 SNPs. rsID#'s of published GWAS SNPs were retrieved for Crea, Cys, and CKD from https://www.nature.com/ng/journal/v42/n5/full/ng.568.html via Table 2.

**ENIGMA22015** [48]: Summary data for ICV, Accumbens, Amygdala, Caudate, Hippocampus, Pallidum, Putamen, and Thalamus were downloaded from http://enigma.ini.usc.edu/research/download-enigma-gwas-results/. We used a minimum $N$ threshold of 10,000, a MAF threshold of 1% and a univariate significant GWAS $p$-value threshold of $5 \times 10^{-8}$. All variants were oriented to the ICV minor allele. The final merged and QC'ed datafile contained 6,271,117 SNPs. rsID#'s of published GWAS SNPs were retrieved for all 8 phenotypes from https://www.nature.com/nature/journal/v520/n7546/full/nature14101.html via Table 1.

**bmass.** bmass implements in an R package the statistical methods described in [14], which should be consulted for full details. In particular, the sections "Computation" and "Detailed Methods (Global Lipids Analysis)" in [14] describe how multivariate analyses are applied to GWAS summary data, and bmass implements the data analysis pipeline described in the "Detailed Methods (Global Lipids Analysis)" section. The bmass R package also includes two vignettes to help users begin to process GWAS summary data and implement these methods.

## Additional details for Fig 3

For each dataset we made a list of "marginally-significant" SNPs, with $p$-values smaller than $1 \times 10^{-6}$ but not genome-wide significant at the relevant datasets' GWAS threshold. We then greedily pruned these lists of marginally-significant SNPs: that is we repeatedly went through the lists removing SNPs that were less significant than another SNP within 500kb. We then removed any SNPs that were within 500kb of a new multivariate association, and merged the resulting list with the list of new multivariate associations, and sorted this merged list of SNPs by their minimum univariate $p$-value.

This results in a non-redundant list of marginally-significant SNPs—some of which are new multivariate associations and some of which are not—sorted by their smallest univariate $p$-value. The plot shows how the number of SNPs of each type varies as the $p$-value threshold is relaxed from the GWAS threshold to $10^{-6}$ (the HaemgenRBC2016 results show only the top 500 SNPs due to the abundance of SNPs between $8.31 \times 10^{-9}$ and $1 \times 10^{-6}$).

## Supporting information

**S1 Fig. Graphical model of multivariate categories.** Shown here is a Directed Acyclic Graphical (DAG) model of our multivariate categories in the context of our vector of phenotypes $\mathbf{Y}$ (e.g. $\mathbf{Y} = \{\mathbf{Y_U}, \mathbf{Y_D}, \mathbf{Y_I}\}$) and their connections with the variant of interest $\mathbf{g}$. The relationships described in-text can be seen here. $\mathbf{Y_U}$, our unassociated phenotypes, have no connection with $\mathbf{g}$. $\mathbf{Y_D}$, our directly associated phenotypes, have a direct connection with $\mathbf{g}$. And $\mathbf{Y_I}$, our indirectly associated phenotypes, have a connection with $\mathbf{g}$ only by going through $\mathbf{Y_D}$ first. Note that if $\mathbf{Y_D}$ were not observed, $\mathbf{Y_I}$ would appear as a direct connection.
(PDF)

**S2 Fig. Refining association signals—GlobalLipids2013 rs7515577 and rs12038699.** Shown are the -$\log_{10}$ univariate *p*-values from the GlobalLipids2013 analysis for both the previous univariate association rs7515577 ("Previous Univariate SNP") and the new multivariate association rs12038699 ("New Multivariate SNP") across all four phenotypes analyzed. rs7515577 is represented as a triangle and rs12038699 is represented as a square. Also shown are the -$\log_{10}$ univariate *p*-values of SNPs within 1Mb of the midpoint between rs7515577 and rs12038699. Color-coding of the SNPs represent the degree of linkage disequilibrium between variants and the new association rs12038699 based on the GBR cohort of 1000Genomes [49]; for color coding details, see legend.
(PDF)

**S3 Fig. Effect size heterogeneity among SNPs with identical multivariate model assignments.** Shown are the phenotype effect sizes (points), and ±2 standard errors (bars), for four significantly associated SNPs from HaemgenRBC2016. All four SNPs were classified as being "associated" with all six phenotypes (i.e. marginal posterior probability of association >= 95% for each phenotype). However, they clearly show different patterns of effect sizes. Therefore focusing simply on binary calls of "associated" vs "unassociated" can hide different patterns of multivariate association.
(PDF)

**S1 Table. Summary of associations in each dataset.** [a] Number of new multivariate associations discovered by our analysis. Note that we required a multivariate association to be at least 500kb from a previous reported association to be considered "new". [b] Univariate GWAS significance *p*-value threshold used by the original study publication. [c] These are new multivariate SNPs that were not reported by the original study despite having a univariate association (in the public summary data) that was genome-wide significant by the original study's univariate significance threshold. [d] A "previous association" means an association reported by the original GWAS; "near" means within 1Mb (but these are all more than 500kb away from a previous association since our classification of new multivariate SNPs requires this).
(PDF)

**S2 Table. Lists of new `bmass` multivariate associations, per dataset.** Attached Excel sheets list new `bmass` associations for each dataset analyzed.
(XLS)

**S3 Table. Lists of retrieved univariate associations from original publications, per dataset.** Attached Excel sheets list the rsID#'s of the univariate significant SNPs that were retrieved from the original publication(s) associated with each dataset (see Methods for details).
(XLS)

**S4 Table. Results for previous univariate associations, per dataset.** Attached Excel sheets give `bmass` results for previous univariate associations, per dataset. Note that these results may not include all SNPs from S3a–S3m Table, because some SNPs were dropped during QC and other SNPs were dropped because they did not reach univariate significance in the publicly available summary data (see Methods for details).
(XLS)

**S5 Table. Replication of new multivariate associations.** Shown are example metrics of how well our new multivariate associations replicate in datasets that allow such an evaluation. Specifically, for three of the studies used (GlobalLipids, GIANT, and HaemgenRBC), there are multiple dataset releases. To examine how well our new multivariate `bmass` associations replicate, we compared the results from the first releases ("1$^{st}$") with the univariate GWAS

associations of the second releases ("2nd"). In essence, each of these approaches aim to increase power—one by using a multivariate approach (bmass) and the other by increasing sample size (the 2nd releases)—thus allowing us to compare the results against one another. Univariate *p*-Value Threshold: univariate GWAS significance *p*-value thresholds used by the original publication(s) for both the earlier (1st) and later (2nd) releases. New Multivariate SNPs in 1st: number of new multivariate associations from the earlier release. Lower Univariate *p*-Value in 2nd: number of new multivariate associations from the earlier release that also have lower univariate *p*-values in the later release. Below 2nd Univariate Threshold: number of new multivariate associations from the earlier release that also cross the later release's univariate GWAS significance threshold.
(PDF)

**S6 Table. *p*-Values for rs7515577 & rs12038699 in 2010 and 2013 GlobalLipids releases.** In the 2010 release rs7515577 has a univariate *p*-value that crosses the $5 \times 10^{-8}$ threshold (TC), whereas rs12038699 does not. Since rs12038699 is near to rs7515577 it may get masked for future analyses; however in the 2013 data rs12038699 not only has a lower minimum univariate *p*-value, but also has a different multivariate *p*-value pattern as compared to rs7515577. Both these signals suggest that rs12038699 should be viewed as a separate GWAS hit for GlobalLipids2013.
(PDF)

**S7 Table. Top multivariate model examples per SNP.** List of multivariate models that most frequently have the highest posterior probabilities per SNP. Top 5 models are shown from across both the previous univariate associations analyzed and the new multivariate associations discovered in the GlobalLipids2013, GIANT2014/5, and HaemgenRBC2016 datasets. Phenotype ordering is shown in the header, where 0, 1, and 2 refer to the multivariate categories of **U**nassociated, **D**irectly Associated, and **I**ndirectly Associated. n is the number of SNPs that show the specified model as having the largest posterior probability, with Mean Posterior displaying the average posterior probability of the given model across the n SNPs, and Original Prior showing the prior established for the given model from training on all the previous univariate associations from that dataset.
(PDF)

## Acknowledgments

We thank John Novembre, Anna Di Rienzo, and Xin He for helpful feedback during the development of this project. We also thank Peter Carbonetto for helpful feedback on the bmass R package and the manuscript.

## Author Contributions

**Conceptualization:** Michael C. Turchin, Matthew Stephens.

**Data curation:** Michael C. Turchin.

**Formal analysis:** Michael C. Turchin.

**Funding acquisition:** Michael C. Turchin, Matthew Stephens.

**Investigation:** Michael C. Turchin.

**Methodology:** Matthew Stephens.

**Project administration:** Matthew Stephens.

**Resources:** Michael C. Turchin, Matthew Stephens.

**Software:** Michael C. Turchin, Matthew Stephens.

**Supervision:** Matthew Stephens.

**Validation:** Michael C. Turchin.

**Visualization:** Michael C. Turchin.

**Writing – Original Draft:** Michael C. Turchin.

**Writing – review & editing:** Michael C. Turchin, Matthew Stephens.

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
