## [Decision Letter · Decision Letter 0]

10 Jul 2019

Dear Dr Turchin,

Thank you very much for submitting your Research Article entitled 'Bayesian multivariate reanalysis of large genetic studies identifies many new associations' to PLOS Genetics. Your manuscript was fully evaluated at the editorial level and by independent peer reviewers. The reviewers appreciated the attention to an important topic but identified some aspects of the manuscript that should be improved.

We therefore ask you to modify the manuscript according to the review recommendations before we can consider your manuscript for acceptance. Your revisions should address the specific points made by each reviewer.

[LINK]

Yours sincerely,

Michael P. Epstein

Associate Editor

PLOS Genetics

Hua Tang

Section Editor: Natural Variation

PLOS Genetics

Reviewer's Responses to Questions

**Comments to the Authors:**

Reviewer #1: Turchin & Stephens show that many more associations can be identified with their Bayesian multivariate analysis. This paper advocates an important message that statistical power can easily improved from univariate to multivariate analysis.

Overall the paper is well written and results are striking. One interesting question is that whether the performance gain is from the bayesian framework (i.e., bmass) or from multivariate analysis, a simple way to address that is to apply another multivariate method and compare its results

with that from bmass.

Reviewer #2: Overview:

This manuscript takes the much needed step to show consortiums what they miss by not considering multivariate analysis of related phenotypes. The authors re-analyzed several publicly available datasets to compare findings from multivariate analysis of multiple phenotypes against those from single-trait analyses. Further, the authors empirically validate their additional findings from multivariate analysis by replicating those findings in larger, more recent datasets (e.g. in studies with multiple releases where later releases have larger sample sizes). The authors additionally emphasize that multivariate analysis is different from multiple single-trait analyses.

I have a few discussion-type comments that will likely help other researchers adopt multivariate analyses in GWAS (which seems to be the goal of this manuscript) and a few minor comments for the authors to consider.

*Major comments*

1. The authors give a couple reasons for why multivariate analyses are not adopted more widely. One reason that I have encountered from consortiums is that two traits may have widely different sample sizes, and no one wants to “sacrifice” higher sample size of one trait by trying to jointly analyze the two traits. This idea is probably stemming from the fact that multivariate analysis based on individual-level data requires the two traits to be measured on the same set of individuals. So, I think it will be useful to emphasize that these days, multivariate analysis can be done using methods based on GWAS summary statistics that do not have such a requirement. However, it also comes with additional questions that needs discussion: 

(a) if the consortiums have individual-level phenotype-genotype data where each trait has widely varying sample sizes, should the analysts first obtain single-trait single-SNP GWAS summary statistics and then proceed with multivariate analysis (using methods based on summary data)? Is this 2-step process an efficient approach? What would the authors recommend? 

(b) are the multivariate methods based on summary data robust across wide differences in sample sizes? Or can the authors give a thumb-rule (that analysts can follow) on the ratio of sample sizes of the traits beyond which summary-data-based-multivariate-methods may not be robust and may not prove more effective than univariate analysis?

  In my opinion, another hurdle in the adoption of multivariate analysis in consortiums: usually the multivariate analyses focus on genetic variants with MAF 1% or more (due to concerns about type I error inflation) while univariate analyses can conveniently go to lower MAFs without as much worry about inflation. Consequently, univariate analyses may end up detecting significantly associated genetic regions that are not even considered for multivariate analyses in the first place (due to more stringent MAF screening). Apparently then multivariate analysis is not giving anything over and above the findings from univariate analysis - the end result is the publication of univariate analysis results only. How would the authors sell multivariate methods to consortiums in such a context where univariate analysis appears more effective?

2. Since the authors are advocating multivariate analyses (which I completely agree with), I think readers will find it useful if the authors can provide a general multivariate analysis pipeline that consortiums can broadly follow. For instance, a general univariate analysis pipeline involves (1) linear or logistic regression of each trait, (2) then clumping of significant variants into unique loci, (3) then performing conditional analyses to determine which signals are independent from signals that are known from past studies, (4) then trying to find biological relevance of independent new signals for the corresponding trait, etc. On the other hand, for multivariate analysis, to avoid the problem of sacrificing higher sample size of one trait (as pointed out in the previous comment), (1) the first step will still be univariate analysis, (2) then multivariate analysis using say BFav approach based on the summary stats from first step (is BFav only applicable to continuous traits?), (3) then clumping of unique loci, (4) then performing conditional analyses? How to do conditional analyses with BFav approach? Which SNP-trait associations from past studies should be considered for conditional analyses? For the 5th step, how should one go about determining biological relevance of a multivariate signal? What next steps can be taken?

3. The authors defined “new multivariate association” as significant “if its BFav exceeds that of any previous univariate association’s BFav in the same study (Stephens, 2013). The rationale is that the evidence for these multivariate associations exceeds the evidence for univariate associations that are generally accepted as likely to be real.” - Is this, in some sense, equivalent to (in the frequentist world) defining significance by looking at the multivariate p-value; if it is smaller than the minimum of the univariate p-values, then we have a new multivariate association?  If so, are we going to call this situation new multivariate association: p_Trait1 = 0.01, p_Trait2=0.005 and p_multivariate=0.001?  If yes, the definition does not seem right because in GWAS, we look at more stringent significance thresholds. Can the authors explain in more detail?

4. Figure 2: Some of the new multivariate SNPs are below the 45-degree line. If I am interpreting it correctly, these SNPs have larger negative log-transformed p-values (i.e., more significant) in the earlier release than the later release (later release with larger sample size). When can this happen? Are these variants low-frequency?

*Minor Comments*

1. Since the authors recommend re-analyzing published GWAS using multivariate methods that use single-SNP summary data, I think it is worth emphasizing that consortiums ensure detailed summary data files are made publicly available. For instance, I have often come across publicly available summary data files that are almost useless because they have missing MAF or risk allele frequency, or missing info on the risk allele or the reference allele, or missing info on the Z statistics (or beta estimates) making direction of effect indeterminable.

2. “in total, 84 of 94 new associations have smaller univariate p-values in the later release, and indeed the majority of these reach univariate GWAS significance in the later release.” - Do the authors mean exactly the same variants have smaller univariate p-values in later release compared to the first release?

3. “at a univariate threshold of 5e-7 only 66% of the univariate significant SNPs are also multivariate significant across these three studies.” - Was 5e-7 threshold used for determining multivariate significance as well? What’s the equivalent BFav value (since the multivariate approach used the BFav Bayesian approach)?

4. Can the authors explain the following choices?

(a) Looks like the author screened our variants with MAF < 1% for all datasets but one. The MAF threshold for GEFOS2015 is 0.5%.

(b) “We used a minimum N threshold of 50,000”.

(c) The authors used “a univariate significant GWAS p-value threshold of 5e-8” for most studies but not all (e.g., for HaemgenRBC2016, threshold used is 8.319e-9).

5. “There may be many reasons why such variants went unreported, but one reason may be physical proximity to a variant with a stronger signal.” - Is it possible that such variants were not reported because they were no longer significant in subsequent conditional analyses?

6. Supplementary Figure 3 plots effect sizes for each trait. Does the BFav approach provide effect size estimates as well?

Reviewer #3: This is a nicely written paper applying a multivariate method to 13 datasets to search for new associations. The paper presents a large number of new findings, with the ultimate goal of encouraging increased use of the multivariate method.

General comments:

The performance of this multivariate approach is illustrated by looking at very closely related phenotypes, such as blood cell subtypes and anthropometric measures. Would this method still yield substantial power gains when investigating joint associations for less similar traits (i.e.: BMI and different cancer types)? Is the power of this approach related to the degree of correlation (genetic and/or phenotypic) between the traits?

The authors note that “we declared a multivariate association as significant if its BFav exceeds that of any previous univariate association’s BFav in the same study.” Thus, the multivariate is deemed significant if it has a Bayes factor larger than any one univariate result. Is this criterion too liberal, and might it result in false positives for the multivariate approach?

It is unclear why examining associations “that would be declared significant if the univariate significance threshold were relaxed” and comparing these to multivariate association results illustrates that multivariate analysis is not the same as multiple univariate analyses. I agree that these are not the same thing, but wouldn’t it make sense to look at whether a given SNP is associated with traits 1-3 in multiple univariate analyses and then see if it is also identified in the multivariate analysis of the same traits?

The “Reanalysis also identifies new univariate associations” part is a little vague. Does this mean that there were significant SNPs in the original summary stats files that were simply not reported? Or are these new SNPs that were identified in the multivariate analysis, but were only associated with one trait?

Showing replication is very important to support the use of the multivariate approach. When using a later release of the same study/consortium to replicate associations, were the older studies excluded from the new release? What is the degree of overlap between the two, and can this be considered independent replication? If the initial study is included in the latter release, wouldn’t one expect the univariate p-value to decrease based on the larger sample size regardless of it being a true signal?

When evaluating the independence of new multivariate associations, it seems that pruning was based on distance (SNPs at least 0.5 Mb apart), but was LD was explicitly considered based on an r2 cut-off?

Why were different significance thresholds used for different studies? HaemgenRBC 2012: p<1�10-8, HaemgenRBC 2016: p<8.319�10-9, GEFOS2015: p<1.2�10-8

In the Reanalysis section the authors state “Indeed the very different multivariate patterns of effect size suggest that not only are these associations non-redundant but likely involve different biological mechanisms as well.” Is there evidence to support this statement? Do we really think that different effect sizes mean different mechanisms?

**Have all data underlying the figures and results presented in the manuscript been provided?**

Reviewer #1: Yes

Reviewer #2: Yes

Reviewer #3: Yes

PLOS authors have the option to publish the peer review history of their article (what does this mean?). If published, this will include your full peer review and any attached files.

Reviewer #1: No

Reviewer #2: No

Reviewer #3: No

---

## [Decision Letter · Decision Letter 1]

4 Sep 2019

[EXSCINDED]

Dear Dr Turchin,

Thank you very much for submitting your revised Research Article entitled 'Bayesian multivariate reanalysis of large genetic studies identifies many new associations' to PLOS Genetics. The peer reviewers were satisfied with most of your revisions, although one reviewer requested some additional minor edits. We therefore ask you to modify the manuscript accordingly before we can move forward with formal acceptance of your work. 

[LINK]

Yours sincerely,

Michael P. Epstein

Associate Editor

PLOS Genetics

Hua Tang

Section Editor: Natural Variation

PLOS Genetics

Reviewer's Responses to Questions

**Comments to the Authors:**

Reviewer #1: My comments have been addressed.

Reviewer #2: I thank the authors for addressing my previous comments. With regards to the authors’ “response to reviewers”, it would have been less time intensive to review if the authors highlighted the text they added in the manuscript. Or, at least mention the relevant page numbers wherever “We have now included […] in the manuscript” is mentioned in their response. I’d request the authors to consider giving this info and/or an annotated manuscript in any subsequent review.

*Minor Comments*

1. The authors added “More generally, although not necessarily essential for our analyses here, it is always helpful to include additional key meta-data (e.g. the reference allele, or effect allele, the minor allele frequency, and sample size).” However, earlier the authors mentioned in section 6.1 that they dropped variants from GWAS datasets that had missing MAF or for which reference/alt alleles did not match across phenotypes. So, I am not sure why the authors say meta-data such as “reference allele, or effect allele, the minor allele frequency” are “not necessarily essential” for their analyses. It may not be necessary for implementing their R program bmass but these info seem essential for subsequent interpretation (without which, there’s high risk of erroneous conclusions about statistical significance - e.g., mixing up effect alleles across traits can inflate results; results may not be trustworthy if the MAF is low/rare).

2. In the response, the authors mention “In particular our current framework does not immediately have options for including covariates in the multivariate analysis, so correcting for confounders such as population structure should be done on the first set of univariate analyses. We have added specific mention of this to the Discussion.” In the Discussion, I found this text: “in many cases summary data remain much easier to obtain and work with; there are big practical advantages as well to modular pipelines that first compute summary data and then use these as inputs to subsequent (more sophisticated) analyses. For example, the multivariate analyses we present here are simplified by assuming that the summary data were computed while adequately adjusting for population stratification.” I may have missed other relevant text but the above quoted text from Discussion does not seem to emphasize/convey the message that any necessary covariate adjustment *has* to be done before obtaining the single-trait GWAS summary statistics, and cannot be done at the multivariate analysis step. So, if one has access to GWAS summary data from a study that did not adjust for population stratification, one cannot expect it to be taken care of by the multivariate analysis.

3. In the response, the authors mention “Under the scenario where a consortium finds that running a multivariate analysis does not indicate any new SNPs as being genome-wide significant, then we would not criticize them for choosing to focus on univariate analyses. We have added a discussion of this to the manuscript.” I have probably missed but I couldn’t find this discussion in the manuscript.

4. It is not clear if the BFav approach is only applicable to continuous traits (if I am not mistaken, the authors have analyzed continuous traits only in this manuscript). If so, it is worth emphasizing.

Reviewer #3: No additional comments.

**Have all data underlying the figures and results presented in the manuscript been provided?**

Reviewer #1: Yes

Reviewer #2: Yes

Reviewer #3: Yes

PLOS authors have the option to publish the peer review history of their article (what does this mean?). If published, this will include your full peer review and any attached files.

Reviewer #1: No

Reviewer #2: No

Reviewer #3: No

---

## [Editor Report · Decision Letter 2]

17 Sep 2019

Dear Dr Turchin,

We are pleased to inform you that your manuscript entitled "Bayesian multivariate reanalysis of large genetic studies identifies many new associations" has been editorially accepted for publication in PLOS Genetics. Congratulations!

Yours sincerely,

Michael P. Epstein

Associate Editor

PLOS Genetics

Hua Tang

Section Editor: Natural Variation

PLOS Genetics

Comments from the reviewers (if applicable):

**Data Deposition**

http://datadryad.org/submit?journalID=pgenetics&manu=PGENETICS-D-19-00888R2

**Press Queries**

---

## [Editor Report · Acceptance letter]

2 Oct 2019

PGENETICS-D-19-00888R2 

Bayesian multivariate reanalysis of large genetic studies identifies many new associations 

Dear Dr Turchin, 

We are pleased to inform you that your manuscript entitled "Bayesian multivariate reanalysis of large genetic studies identifies many new associations" has been formally accepted for publication in PLOS Genetics! Your manuscript is now with our production department and you will be notified of the publication date in due course.

With kind regards,

Kaitlin Butler

PLOS Genetics

On behalf of:
